# A nanobody-based fluorescent reporter reveals human α-synuclein in the cell cytosol

Christoph Gerdes [1], Natalia Waal [2], Thomas Offner[3,4], Eugenio F. Fornasiero [1], Nora Wender[1], Hannes Verbarg[1], Ivan Manzini [3,4], Claudia Trenkwalder[5,6], Brit Mollenhauer[6,7], Timo Strohäker[8], Markus Zweckstetter[8,9], Stefan Becker [9], Silvio O. Rizzoli[1,10], Fitnat Buket Basmanav[1,11,12] & Felipe Opazo [1,2,12 ✉]

Aggregation and spreading of α-Synuclein (αSyn) are hallmarks of several neurodegenerative diseases, thus monitoring human αSyn (hαSyn) in animal models or cell cultures is vital for the field. However, the detection of native hαSyn in such systems is challenging. We show that the nanobody NbSyn87, previously-described to bind hαSyn, also shows cross-reactivity for the proteasomal subunit Rpn10. As such, when the NbSyn87 is expressed in the absence of hαSyn, it is continuously degraded by the proteasome, while it is stabilized when it binds to hαSyn. Here, we exploit this feature to design a new Fluorescent Reporter for hαSyn (FluoReSyn) by fusing NbSyn87 to fluorescent proteins, which results in fluorescence signal fluctuations depending on the presence and amounts of intracellular hαSyn. We characterize this biosensor in cells and tissues to finally reveal the presence of transmittable αSyn in human cerebrospinal fluid, demonstrating the potential of FluoReSyn for clinical research and diagnostics.

[1] Department of Neuro- and Sensory Physiology, University Medical Center Göttingen, D-37073 Göttingen, Germany. [2] Center for Biostructural Imaging of Neurodegeneration (BIN), University Medical Center Göttingen, D-37073 Göttingen, Germany. [3] Institute of Animal Physiology, Department of Animal Physiology and Molecular Biomedicine, Justus-Liebig University Giessen, 35390 Giessen, Germany. [4] Institute of Neurophysiology and Cellular Biophysics, University of Göttingen, Göttingen, Germany. [5] Department of Neurosurgery, University Medical Center Göttingen, D-37075 Göttingen, Germany. [6] Paracelsus-Elena-Klinik, Klinikstraße 16, 34128 Kassel, Germany. [7] Department of Neurology, University Medical Center Göttingen, D-37075 Göttingen, Germany. [8] German Center for Neurodegenerative Diseases (DZNE), Von-Siebold-Str. 3a, 37075 Göttingen, Germany. [9] Department for NMR-based Structural Biology, Max Planck Institute for Biophysical Chemistry, 37077 Göttingen, Germany. [10] Cluster of Excellence "Multiscale Bioimaging: from Molecular Machines to Networks of Excitable Cells" (MBExC), University of Goettingen, Göttingen, Germany. [11] Campus Laboratory for Advanced Imaging, Microscopy and Spectroscopy, University of Göttingen, D-37073 Göttingen, Germany. [12] These authors contributed equally: Fitnat Buket Basmanav, Felipe Opazo. ✉email: fopazo@gwdg.de

α-synuclein (αSyn) aggregation disorders including Parkinson's disease (PD), Lewy Body dementia, and multiple system atrophy, are a group of disorders characterized by the pathological occurrence of intracellular inclusions filled with insoluble aggregates of αSyn. These aggregations can be found in various cell types and regions of the central nervous system[1]. αSyn is a 140 amino-acid long protein that is highly enriched in presynaptic nerve terminals[2]. Despite extensive efforts, the central molecular and physiological role of αSyn remains to be determined. Emerging evidence, however, suggests αSyn to be involved in the regulation and possible maturation of synaptic vesicles[3], particularly via its role in the assembly of N-ethylmaleimide-sensitive factor attachment receptor complexes[4]; key players in synaptic vesicle docking and fusion with the presynaptic membrane. In addition, other functions have been attributed to αSyn including the regulation of glucose levels[5], serving as an antioxidant[6] or a chaperone[7], and suppressing apoptosis in dopaminergic neurons[8].

Under normal physiological conditions, αSyn is found as a monomeric protein existing in equilibrium between an intrinsically disordered form in the cytosol and a membrane bound, α-helical form[9]. Although still subjected to debate[10], observation of helical αSyn tetramers has also been reported under certain native environments[11].

It is widely accepted that arrangements of αSyn monomers into small-to-intermediate oligomeric or larger insoluble assemblies[12] is associated with pathogenesis of αSyn aggregation disorders[13]. Importantly, there is growing evidence for prion-like cell-to-cell transmission properties of different αSyn conformations where "toxic" αSyn species with seeding properties are suggested to get internalized by a host cell and trigger the aggregation of endogenous αSyn[14–18]. There are nevertheless several assumptions, unknown molecular steps and contradictory observations that render the toxic transmission of human αSyn (hαSyn) an ambiguous notion. These mainly concern pinpointing of the disease underlying αSyn species (e.g., oligomers vs fibrils, phosphorylated αSyn, etc)[19], mechanistic steps of the cell-to-cell transmission paradigm[10,20] and the downstream effects of pathological αSyn accumulation that eventually lead to neuronal injury[21].

Development of tools that can be used for reliable and reproducible detection and tracking of αSyn in in vitro or in vivo is a very important goal for deciphering molecular mechanisms of disease pathogenesis. A common strategy employed in studies addressing the cellular uptake, seeding, and transmission phenomena involve manipulation of the αSyn protein itself. Accordingly, αSyn is either pre-labeled with fluorescent dyes[15,22,23], or recombinantly expressed as a fusion protein. Concerning the latter, several different approaches are described which include tagging αSyn with a small epitope tag (e.g., myc, V5, etc.) for subsequent antibody-mediated immunodetection[15,24], expressing it as a fluorescent fusion protein (e.g., YFP-αSyn, DsRed-αSyn, etc.) for direct visualization[25,26] or employing the protein complementation assay principles whereby αSyn is tagged with either N-terminal or C-terminal portions of a split fluorescent or bioluminescent reporter[27]. Although they have proven to be very helpful in advancing the knowledge about synuclein-related pathologies, one common caveat in such approaches is their limited potential in recapitulating the natural behavior of untagged native αSyn. For example, it has been shown that fluorescent protein fusions might result in the wrong localization of the studied protein[28] or some organic dyes have a propensity to bind to biological membranes[29]. Considering that a critical role is attributed to the lipid interacting properties of αSyn in its pathological behavior[30], the employment of untagged or native forms of αSyn may be a more-appropriate strategy when attempting to investigate the molecular mechanisms of αSyn pathology. The detection of untagged, native, or endogenous αSyn in in vitro or in vivo model systems requires the use of other tools. Certain dyes such as Thioflavin S, derived from the histological dye Congo red, have been commonly utilized for detecting mature protein aggregates in in vivo models of αSyn propagation and PD pathology[16,31]. However, these dyes have the disadvantage of binding any protein capable of taking an amyloid conformation and thus do not provide an exclusive labeling of αSyn[32]. In contrast, antibody-mediated immunodetection is a conventional approach for specific detection of αSyn. Hereby, a large number of αSyn-targeting antibodies, including conformation specific-ones and engineered antibody fragments are commonly utilized in a variety of applications[33,34]. In the recent years, camelid-originated single-domain antibodies, also termed nanobodies, emerged as a promising alternative as they confer several advantages including recombinant production, enhanced tissue penetration, small size (ideal for super-resolution microscopy), and the ability to be expressed as intrabodies in mammalian cells[35–37]. The latter feature is particularly attractive as it confers the ability to track and manipulate specific target proteins in living cells[38]. Nanobodies are increasingly being used for investigation of diseases associated with protein misfolding and aggregation. Recently, two nanobodies against αSyn have been identified, NbSyn2 (ref. [39]) and NbSyn87 (ref. [40]), each binding distinct epitopes at the C-terminal region of αSyn. These have been biochemically well characterized and assessed for their potential therapeutic use by several studies[39–44].

In this study, we make use of a previously unknown feature of NbSyn87, namely its weak affinity to the 26 S proteasomal subunit protein Rpn10, which is located at the entrance of the proteasome[45] and functions as a receptor for poly-ubiquitinated proteins that will undergo proteolysis. We show that this interaction is sufficient to drive a continuous proteasome-mediated degradation of intracellularly expressed NbSyn87 unless it is bound to hαSyn. The presence of hαSyn, on the other hand, results in the avoidance of the degradation of NbSyn87 by formation of a stabilized NbSyn87:hαSyn complex. Accordingly, we exploit this mechanism to create and characterize a nanobody-based Fluorescent Reporter for human αSyn (FluoReSyn), which is able to report the presence or absence of cytosolic hαSyn. Our results demonstrate the unique ability of FluoReSyn to report small amounts of cytosolic hαSyn in cell lines and transduced primary rat hippocampal neurons. Expression of FluoReSyn in olfactory system of Xenopus laevis also shows its ability to operate and report hαSyn in vivo. Furthermore, cells stably expressing FluoReSyn (Reporter-cells) report the presence of hαSyn in their cytoplasm after exposing them to human cerebrospinal fluid (CSF) samples. The results presented here indicate that this biosensor is a valuable instrument for studying the transmission of αSyn and has great potential to be further optimized and validated as a diagnostic tool for αSyn aggregation disorders.

## Results

**Reporting the presence of untagged hαSyn in the cytoplasm.** We had previously observed in cells transiently expressing the NbSyn87 (ref. [40]) fused to EGFP that their fluorescent signal correlated with the presence or absence of hαSyn (Fig. 1a). In an attempt to comprehend this observation, we used the Basic Local Alignment Research Tool (BLAST) to find out if the described hαSyn epitope sequence (VDPDNEAYEMPS)[40] that is recognized by the NbSyn87 might be present in another endogenous protein. The BLAST result showed a high % identity (Fig. 1b) to a subunit of the 26 S proteasome (the 26 S proteasome non-ATPase regulatory subunit four homolog, also known as Rpn10). This

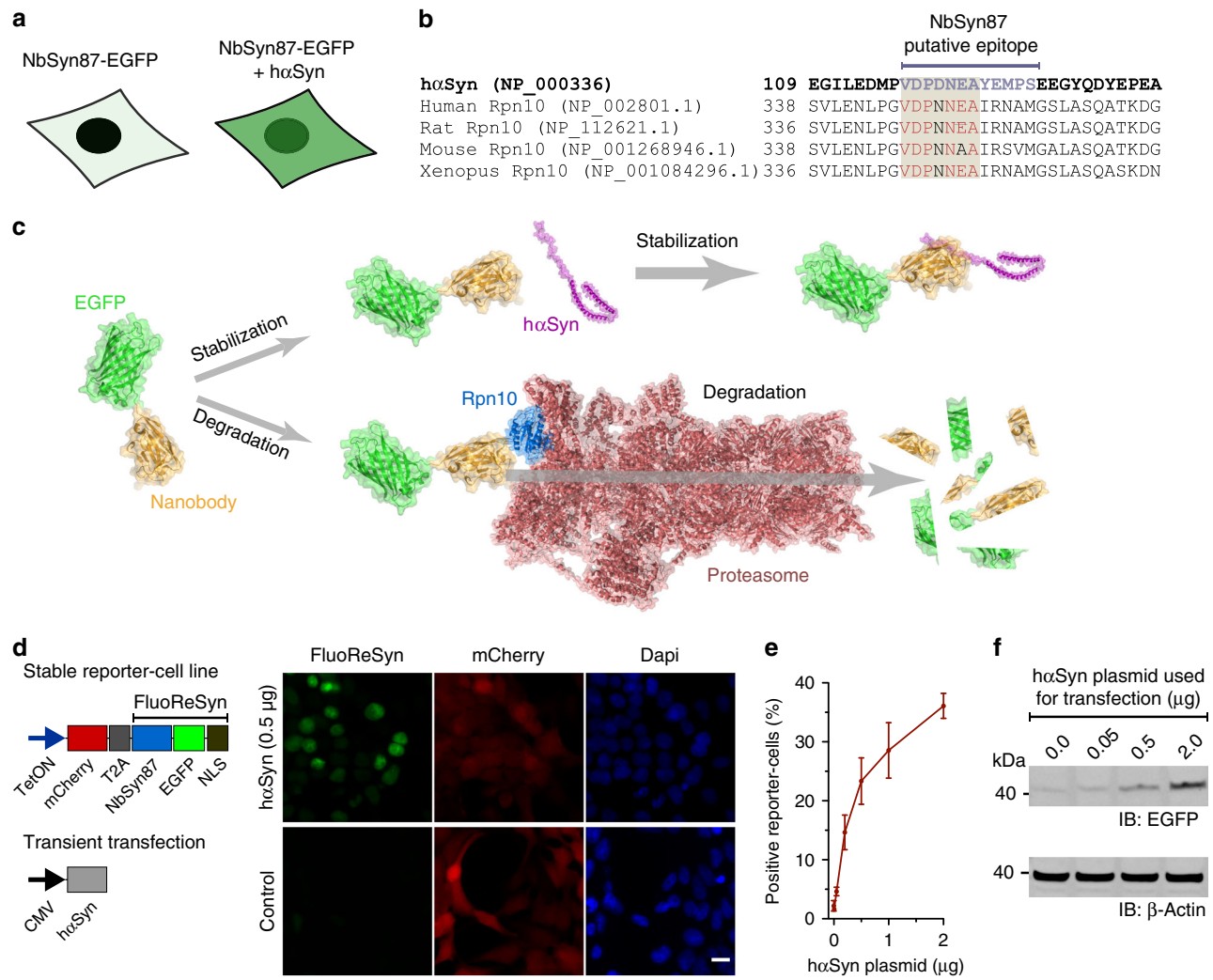

**Fig. 1 αSyn-dependent accumulation of FluoReSyn in HEK293 cells. a** Schematic representation of the initial observation of cells transiently expressing NbSyn87-EGFP alone (left) or together with hαSyn (right). Although the former group of cells showed minimal fluorescence, the latter presented with a strong EGFP signal. **b** Alignment of the amino-acid sequence from hαSyn and Rpn10 across different species. Rpn10 residues similar to hαSyn in the putative epitope are displayed in red. **c** Schematic representation of the proposed mechanism of degradation versus stabilization of SynNb87-EGFP in the presence or absence of hαSyn. The following Protein Data Bank (PDB) accession numbers were used and modified to assemble the schematic: 2Y0G (EGFP), 2 × 6 M (Nanobody), 1XQ8 (hαSyn), 6MSK (Proteasome). **d** Schemes of constructs used to transfect HEK293 cells (left). For the stably transfected cells, a tetracycline-inducible promotor (TetON) was used, followed by a mCherry reporter sequence, a cleavable T2A sequence, and FluoReSyn made of NbSyn87, EGFP and a nuclear localization signal (NLS) sequence. Transient expression of untagged wild-type human hαSyn was driven by a plasmid containing a cytomegalovirus (CMV) promotor. Equally scaled, representative images of doxycycline-induced Reporter-cells (right). Cells were either mock transfected (control) or transiently transfected with the hαSyn expression constructs. Scale bar represents 10 μm **e** Quantitative analysis of EGFP-positive cells transfected with variable quantities of hαSyn plasmid. Per replication and condition >1000 cells were analyzed. Error bars represent the SEM from three independent experiments ($n = 3$). **f** Western blot analysis of lysates from Reporter-cells transfected with variable quantities of hαSyn. Immunoblotted (IB) anti-EGFP represents FluoReSyn. Loading control is IB Beta-Actin. Full length blots are displayed in Supplementary Fig. 1d. Source data is available as a Source data file.

protein resides at the entrance of the 26 S proteasome[45] and has an important role in the recognition of poly-ubiquitinated proteins that will be processed in the ubiquitin proteasome-mediated proteolysis (UPP)[46]. Using a dot-blot assay with purified Rpn10 and hαSyn, we were able to verify that the NbSyn87 can bind weakly to human Rpn10 (Supplementary Fig. 1a, b). Therefore, taking our results together, we hypothesized that the degradation of NbSyn87 in the absence of hαSyn may be mediated by its weak but continuous recruitment to the proteasome upon binding the endogenous Rpn10 (Fig. 1c).

In order to test our hypothesis and further characterize NbSyn87 in terms of this special feature, we decided to generate a stable cell line expressing NbSyn87 fused to EGFP and have

mCherry signal as an expression reporter using the self-cleavable domain T2A[47] (Fig. 1d). In addition, we added a NLS sequence at the C-terminus of NbSyn87-EGFP to concentrate the EGFP signal in the nucleus and gain sensitivity during imaging. The expression of this protein-chimera was controlled under the tetracycline-inducible promotor system (TetOn). Optimal induction duration to maximize the NbSyn87-EGFP-NLS expression was determined by analyzing the mCherry reporter signal (Supplementary Fig. 1c). In line with our original observations, we detected a clear nuclear EGFP signal in doxycycline-induced NbSyn87-EGFP-NLS stable cell line when we transiently transfected them with wild type and untagged hαSyn (Fig. 1d). The strength of this effect was dependent on the amount of hαSyn

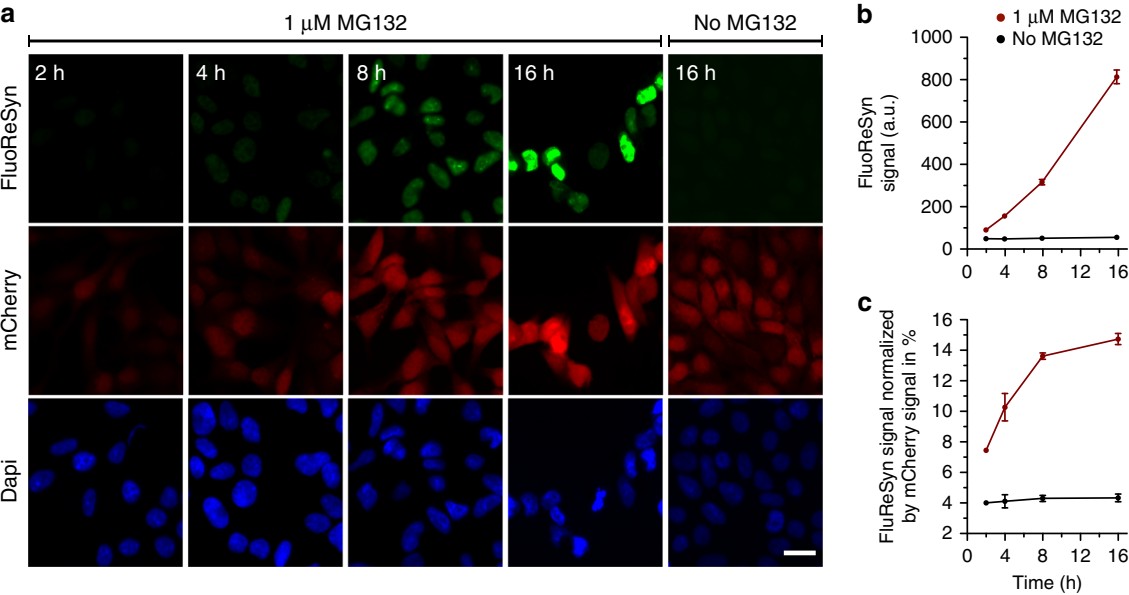

**Fig. 2 The FluoReSyn is rapidly degraded by the proteasome machinery. a** Equally scaled, representative epifluorescence images of MG132 treated and untreated Reporter-cells which were fixed 2, 4, 8, or 16 h post treatment. The FluoReSyn accumulates in response to prolonged MG132 treatment of Reporter-cells as revealed by the increasing nuclear EGFP intensity. Reporter-cells untreated with MG132 yielded virtually no EGFP signal even after 16 h. Scale bar represents 20 μm. **b** Quantitative analyses of FluoReSyn signal (EGFP) in arbitrary units (a.u.) when Reporter-cells were treated or untreated with MG132. Per replication and condition more than 540 cells were analyzed. **c** FluoReSyn signal normalized by mCherry signal (EGFP to mCherry signal intensity). Error bars represent the SEM from three independent experiments ($n = 3$). Source data is available as a Source data file.

present in the cells; detected both by EGFP fluorescence signal intensity analysis (Fig. 1e) and EGFP amounts revealed in western blots (Fig. 1f and Supplementary Fig. 1d). Doxycycline-induced cells that were not transiently transfected with hαSyn displayed the mCherry reporter signal but no EGFP signal (Fig. 1d). Therefore, we termed the NbSyn87-EGFP-NLS construct as the Fluorescent Reporter for αSyn (FluoReSyn) and the stably FluoReSyn expressing cell line as the Reporter-cells.

**Proteasome-mediated degradation of FluoReSyn.** In order to validate our proposed mechanism for the degradation of FluoR-eSyn, we treated doxycycline-induced Reporter-cells (without expressing hαSyn) over different time periods with the MG132 proteasome inhibitor. MG132 concentration was optimized to provide a strong gain of EGFP and mCherry signal while mini-mizing its adverse toxic effects on the Reporter-cells (Supple-mentary Fig. 2). The results clearly showed that, following treatment with 1 μM of MG132, FluoReSyn started to accumulate in the nuclei of Reporter-cells already from the 4th h on without the presence of hαSyn (Fig. 2a, b). Induced Reporter-cells untreated with MG132, showed in contrast virtually no FluoR-eSyn signal throughout the whole duration (Fig. 2a, b). These suggested that the reporter is regularly produced but also con-tinuously degraded via the proteasome in the cell (fast turnover) under normal conditions. The mCherry and FluoReSyn are produced from a single mRNA, making a fusion protein that is efficiently cleaved at the T2A domain[47]. It is expected that this strategy results in stoichiometric amounts of FluoReSyn and mCherry, which allowed us to normalize the signal of FluoReSyn with the mCherry signal. In induced Reporter-cells untreated with MG132, the relation of FluoReSyn to mCherry signal was maintained during the 16 h experiments (Fig. 2c). However, when using MG132, already after 2 h, the ratio between FluoReSyn signal and mCherry doubled and kept growing over time, showing that the accumulation of FluoReSyn exceeds that of mCherry and that the former gets particularly enriched in the

Reporter-cells when the proteasome machinery is inhibited (Fig. 2c). This observation suggests that the proteasomal degra-dation is particularly accelerated for FluoReSyn under normal conditions (uninhibited proteasome machinery), and thus sub-stantiates our proposition that the nanobody NbSyn87 specifically targets the proteasome and is degraded by it.

**FluoReSyn reports hαSyn in vivo.** As Rpn10 is a well-conserved protein across different species (Fig. 1b), we presumed that our proposed mechanism should also operate in a model organism that lacks endogenous hαSyn and possesses the conserved Rpn10 epitope recognized by NbSyn87. Accordingly, we chose to validate the proposed mechanism in living *Xenopus laevis* tadpoles, a time- and cost-efficient model organism[48]. *Xenopus laevis* expresses endogenously the same Rpn10 epitope needed for FluoReSyn to operate and offers a straightforward electroporation-mediated approach for gene delivery to the olfactory receptor neurons in the olfactory epithelium of living animals[49] (Fig. 3a, b). Accordingly, the plasmid encoding for FluoReSyn was electroporated either alone or together with a plasmid encoding for hαSyn fused to mCherry into the right or left nostrils of anesthetized tadpoles, respectively. By in vivo imaging of tadpoles with two-photon microscopy we observed many GFP-positive nuclei co-localizing with the mCherry sig-nal (hASyn) in the left nostrils of the animals (Fig. 3c, e, g, high magnification example on 3j), which was clearly in contrast with the right nostrils presenting seldom any GFP-positive nucleus (Fig. 3d, f, h). Analysis of the distribution of GFP fluorescence intensity further confirmed the clear distinction between left and right nostrils (Fig. 3i). Altogether, these data further validated our previous conclusions by demonstrating that the same mechanism seems to be operational in vivo whereby FluoReSyn is stabilized upon hαSyn binding and cleared from the cell by proteasome-mediated degradation in the absence of this interaction.

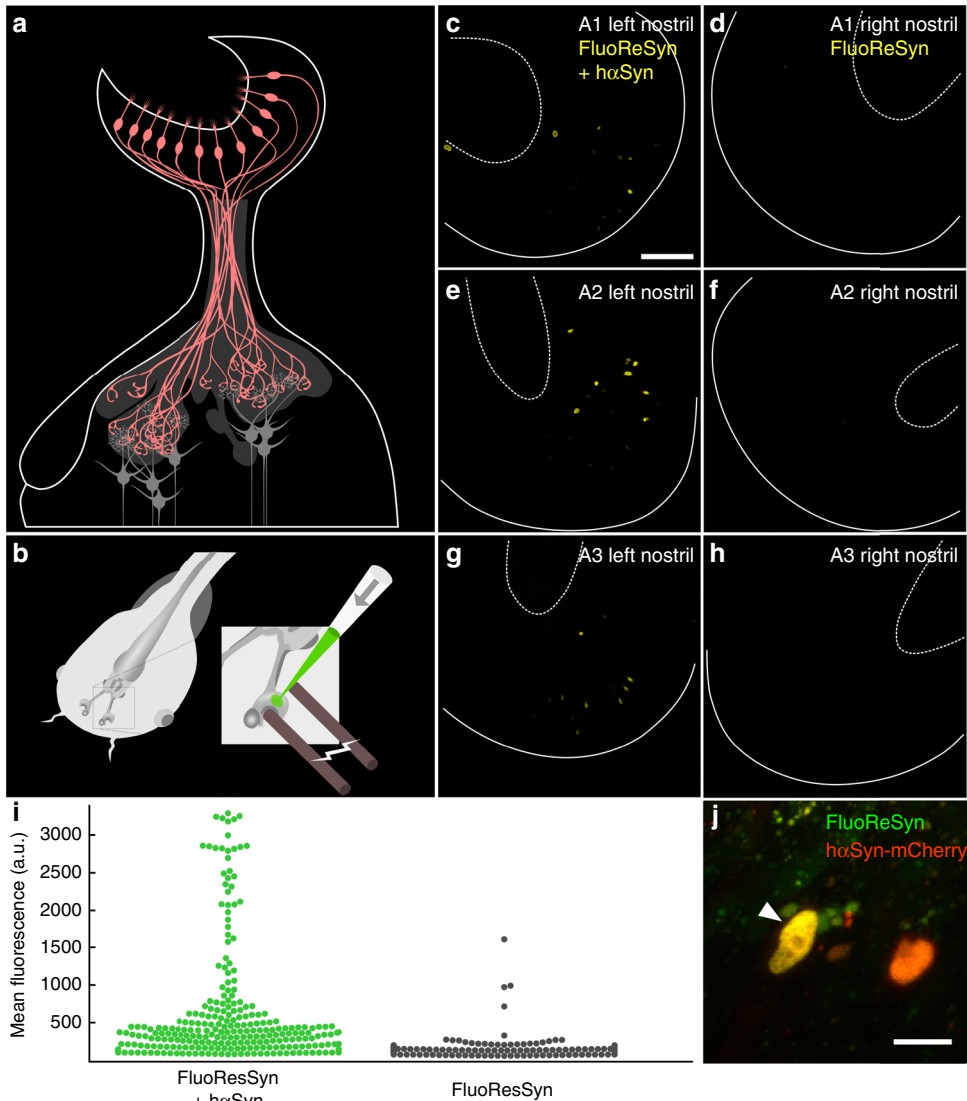

**Fig. 3 in vivo FluoReSyn activity in neurons of larval *Xenopus laevis*. a** Organization of the *Xenopus laevis* olfactory system: olfactory receptor neurons (ORNs; red) residing in the olfactory epithelium project their axons to the olfactory bulb, where they synapse onto the second order projection neurons (gray). **b** Schematic of plasmid electroporation into the olfactory mucosa of anaesthetized *Xenopus laevis* tadpoles. **c**, **e**, **g** Left nostrils of three animals (A1–A3) were co-electroporated with FluoReSyn- and hαSyn-expressing plasmid. Scale bar represents 100 μm. **d**, **f**, **h** Respective right nostrils electroporated only with FluoReSyn. For all animals and nostrils identical plasmid concentrations and imaging settings were used. Basal and apical delineations of olfactory epithelium are indicated by line and dashed line, respectively. **i** Distribution of FluoReSyn mean fluorescence intensity of positive nuclei from the left nostrils (FluoReSyn and hαSyn plasmid; green dots) and nuclei from the right nostrils (FluoReSyn plasmid only; black dots) in arbitrary units (a.u.). More than 500 nuclei were analyzed from nine animals ($n = 9$) imaged in vivo. Unpaired and two tailed student *t* test results in $p < 0.0001$ (****). **j** High magnification example of a neuron co-expressing FluoReSyn (green) and hαSyn-mCherry (red), resulting in nuclear colocalization (yellow) of both proteins. Scale bar represents 10 μm. Source data is available as a Source data file.

**The specificity of the FluoReSyn for hαSyn**. As a next step, we assessed whether FluoReSyn binds specifically to human αSyn or has an affinity towards other synuclein species. For this purpose, Reporter-cells were transiently transfected with plasmids encoding for hαSyn, human ßSyn (hßSyn), or rat αSyn (rαSyn). We analyzed the proportion of cells with a positive EGFP signal by epifluorescence microscopy (Fig. 4). The expressions of the different synuclein species were controlled by immunostaining with an antibody that recognizes all transfected variants. Although no significant differences were observed among the different synuclein species in terms of their expression (Supplementary Fig. 3a), only in hαSyn-transfected Reporter-cells a substantial proportion of the cells were positive for the FluoReSyn signal (Fig. 4b). This result, therefore, proposes that FluoReSyn binds primarily to

human αSyn. It is important to note that the epitope sequence of the hαSyn recognized by NbSny87 has a great identity with the rαSyn sequence with the epitopes differing from each other only by two residues (Supplementary Fig. 3b). It is noteworthy, that FluoReSyn seems to bind with higher affinity to Rpn10 than to the rαSyn.

**Detection of hαSyn uptake from the culture medium**. In relation to the pathological αSyn transmission phenomenon[10], we evaluated the ability of FluoReSyn to report the entry of foreign hαSyn into the cellular cytosol. For this purpose, purified recombinant hαSyn was administered to the culture mediums of induced Reporter-cells either on its own or in a mixed state with a

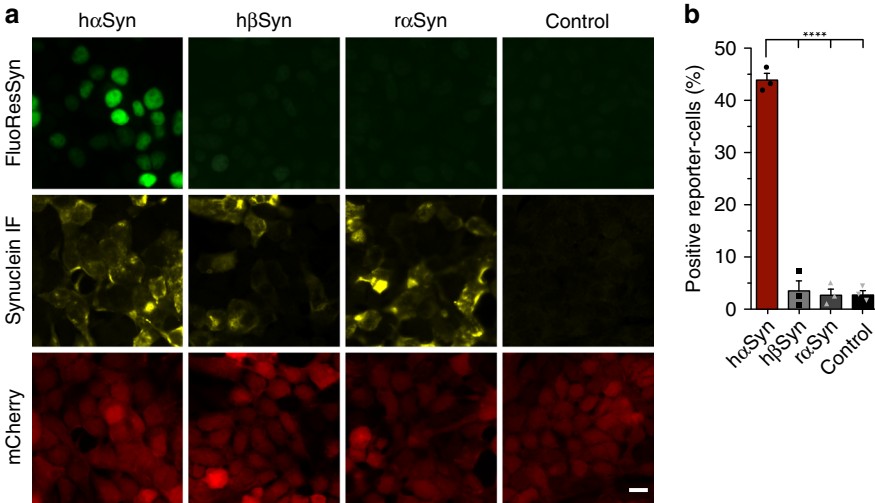

**Fig. 4 The FluoReSyn is a specific sensor for human αSyn. a** Equally scaled representative epifluorescence images of Reporter-cells transfected with hαSyn, hβSyn, or rαSyn expression constructs. Scale bar represents 10 μm. **b** Quantification of the proportion of EGFP-positive cells. Error bars represent the SEM from the independent experiments ($n = 3$). Per replication and condition >6000 cells were analyzed. Ordinary one-way ANOVA resulted in ****$p < 0.0001$. Source data is available as a Source data file.

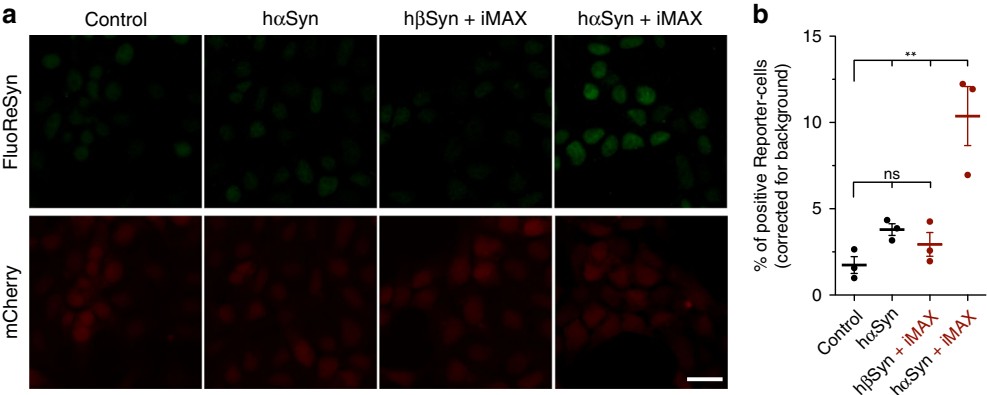

**Fig. 5 Transmission of recombinant hαSyn into Reporter-cells. a** Equally scaled representative epifluorescence images of Reporter-cells after incubation for 14 h with recombinant hαSyn monomers added to the culture medium. Monomers were either administered alone (hαSyn) or pre-mixed with the commercial cationic lipid mixture RNAiMAX (hαSyn + iMAX). hβSyn + iMAX was used as negative control. Scale bar represents 20 μm. **b** Quantification of the percentage of EGFP-positive Reporter-cells. Error bars represent the SEM from three independent experiments ($n = 3$). Per replication and condition >2300 cells were analyzed. One-way ANOVA with multiple comparison Tukey's post hoc test **$p < 0.01$, ns: not significant. Source data is available as a Source data file.

cationic liposome reagent (i.e., RNAiMAX) to enhance the protein uptake as suggested previously[50,51]. We observed that, if hαSyn was associated with RNAiMAX, it manages to get into the cytosol and stabilize FluoReSyn as evidenced by the accumulated EGFP signal in the nucleus (Fig. 5). On the other hand, administrating hβSyn associated with RNAiMAX or just naked hαSyn generated only background levels (i.e., induced Reporter-cells not exposed to anything) of EGFP-positive cells, which suggests that hαSyn on its own failed to go across the cell plasma membrane efficiently.

Subsequently, we setup a co-culture of Reporter-cells and HEK293 cells stably expressing untagged and wild-type hαSyn under an inducible promotor (TetON-hαSyn cells; Supplementary Fig. 4) to investigate if mammalian produced hαSyn was able to leave the cells and then enter into the neighboring Reporter-cells. As a control condition, Reporter-cells were co-cultured with wild-type HEK293 cells, which lack endogenous hαSyn expression. The analysis of co-cultures maintained for 2–5 days revealed

no significant differences between the test and control groups failing to confirm the occurrence of any hαSyn transmission events between the hαSyn producing cells and the Reporter-cells (Supplementary Fig. 5).

**Uptake of hαSyn from the culture medium by primary neurons**. As a next step, we assessed the functionality of FluoReSyn in neurons, by investigating whether it can report the cytosolic presence of hαSyn, after adding recombinant hαSyn to the medium of primary neuronal cultures. For this purpose, we prepared rat hippocampal neuron cultures and infected them at DIV ~14 with an adeno-associated virus (AAV) encoding for NbSyn87-mCherry-NLS (a red version of FluoReSyn). We administered both the monomeric and large fibrillar[52] forms of hαSyn extracellularly to the culture medium of FluoReSyn-transduced hippocampal neurons. The analysis suggested that naked monomeric hαSyn can reach the cytosol of neurons and can produce a detectable FluoReSyn signal in their nuclei

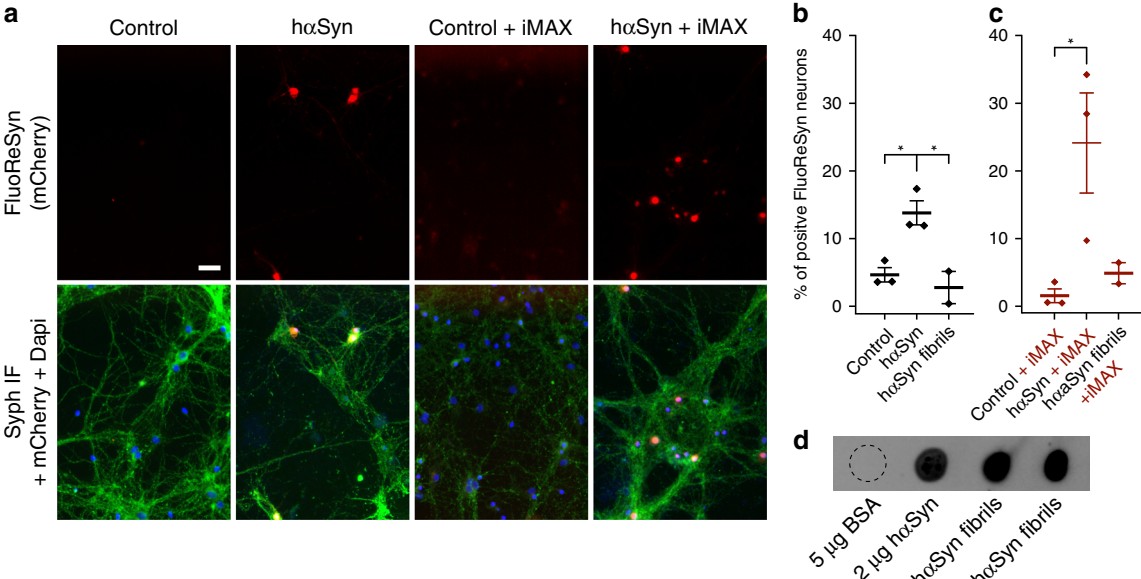

**Fig. 6 Detection of hαSyn monomers added to the medium of cultured neurons. a** Equally scaled, representative images of primary rat hippocampal neurons infected with adeno-associated virus (AAV) encoding for NbSyn87-mCherry-NLS (a red version of the FluoReSyn). The cultures were then exposed to 20 μM of monomeric hαSyn, with or without RNAiMAX (iMAX), for 14 h. Immunofluorescence against synaptophysin was used as a neuronal marker (Syph IF, green signal). The negative controls were neurons receiving Opti-MEM with or without RNAiMAX. Scale bar represents 20 μm. **b, c** Quantification of FluoReSyn signal-positive neurons. One-way ANOVA with multiple comparison Tukey test *$p < 0.05$. Error bars represent the SEM, each point in the scatter plot represents the average of an independent experiment **d** Monomeric untagged hαSyn and two different batches of in vitro produced hαSyn fibrils were spotted on a nitrocellulose membrane and were detected by NbSyn87-Alexa647, which confirmed the ability of NbSyn87 to bind the fibrils used in hippocampal cultures. Bovine serum albumin (BSA) was used as negative control. Source data is available as a Source data file.

(Fig. 6a, b). This was in contrast to the large hαSyn fibrils, which failed to induce FluoReSyn signals above that of the control neurons (Fig. 6b).

Based on our previous observations with the Reporter-cell line, where we also administered the synuclein species pre-mixed with the cationic lipid RNAiMAX (Fig. 6a, c). The results revealed a higher proportion of neurons with FluoReSyn positive nuclei when hαSyn monomers were pre-mixed with the cationic lipids from RNAiMAX and added to the culture medium (Fig. 6a, c). It was interesting to observe that both, the naked monomers (Fig. 6b) and those complexed with lipids (Fig. 6c) were able to go across the neuron plasma membranes. This is clearly distinctive to the HEK293-based Reporter-cells, which displayed internalization of hαSyn only when complexed to cationic lipids (Fig. 5b).

The use of cationic lipids did not influence the outcome for fibrils. Similar to the previous observation with uncoated fibrils (Fig. 6b), no FluoReSyn signal was detected in neurons exposed to RNAiMAX-hαSyn fibril complexes (Fig. 6c). Importantly, we showed that NbSyn87 can clearly bind to our in vitro generated fibrils (Fig. 6d), which also confirmed previous reports[40]. Thereby, we can exclude that the negative observations were owing to the inability of FluoReSyn to detect our fibrils (Fig. 6d), but are most likely a proof that fibrils have not succeeded in entering the cytoplasm of neurons.

**Detection of hαSyn in CSF samples**. Confident that FluoReSyn can reliably report cytosolic hαSyn, we decided to evaluate the ability of the Reporter-cells to detect hαSyn species in human-originated biological samples. The main rationale here was to explore the potential usability of this cellular system for future diagnostic purposes. Therefore, we exposed induced Reporter-cells cultured on a 96-well plate to CSF samples from 42 individuals diagnosed with variable neurological disorders that were

unrelated to αSyn aggregation disorders. Reporter-cells not exposed to CSF, as well as wild-type HEK293 cells exposed and not exposed to CSF, were used as negative controls. Cells were fixed 24 h post treatment and dozens of images of randomized and non-overlapping locations were automatically acquired per well (Fig. 7a). Data analysis showed a small but clear trend of positive Reporter-cells that were incubated with human CSF (Fig. 7b). We also correlated the specific clinical diagnosis of each patient and the total αSyn concentrations in their CSF to the Reporter-cells activity, as displayed in Supplementary Fig. 6. As controls, Reporter-cells not exposed to CSF or wild-type HEK293 cells (not producing any FluoReSyn chimera) incubated with or without CSF, all displayed background levels of positive signal (Fig. 7b). This observation reassured that the small percentage of positive Reporter-cells observed upon CSF exposure is a specific and trustworthy response to some forms of hαSyn present in human CSF that can reach the cytosol of our model Reporter-cells. Altogether, this result not only suggests that human CSF may contain a transmittable form of αSyn that is capable of entering into cells but also opens the possibility to optimize this system for generating a unique cell-based diagnostic tool for αSyn aggregation disorders.

## Discussion

Here, we present for the first time a unique feature of the NbSyn87, namely, its natural tendency to bind to the proteasomal subunit Rpn10 that leads to its own degradation and eventual clearance from the cell cytoplasm in the absence of hαSyn. This special feature allowed us to develop FluoReSyn, the nanobody-based fluorescent reporter for hαSyn, which is capable of detecting the presence or absence of hαSyn in the cellular cytoplasm. Furthermore, our Reporter-cells stably expressing FluoReSyn were able to detect a transmittable form of hαSyn present in human CSF.

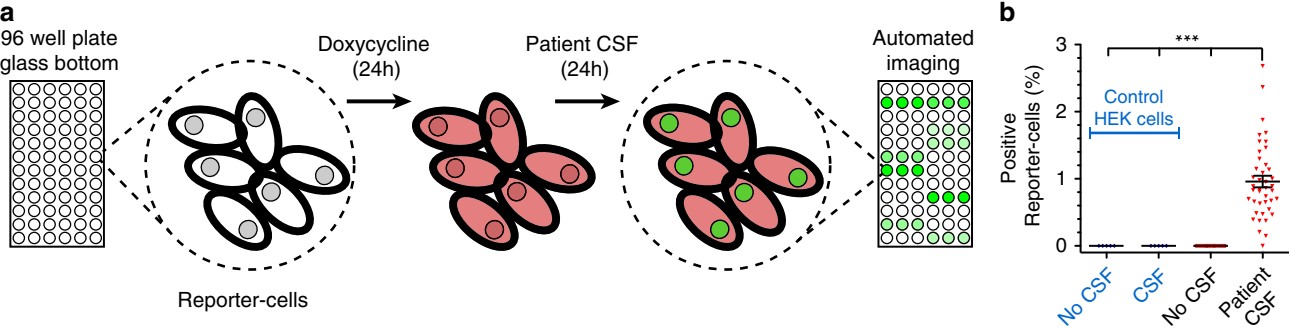

**Fig. 7 Detection of transmittable hαSyn from human CSF samples. a** Illustration of the experimental setup. Reporter-cells or wild-type HEK293 cells (control) were seeded onto glass bottom 96-well plates. FluoReSyn expression was induced by doxycycline for 24 h before they were exposed to human CSF or culture medium (control) for another 24 h. Finally, cells were fixed and ~20 images of random and not overlapping locations were acquired automatically per well. The CSF from each patient was used at least in three different wells. **b** Quantification of cells with positive FluoReSyn signal. In total, results represent data from 126 wells of Reporter-cells receiving CSF samples from 42 individuals in technical triplicates, 102 negative control wells with Reporter-cells receiving culture medium, 15 wells of HEK293 cells receiving CSF samples from five individuals in technical triplicates and 36 wells of HEK293 cells receiving culture medium. One-way ANOVA with multiple comparison Tukey test ***$p < 0.001$. Further correlations between the Reporter-cell activity and the clinical information of each patient are displayed in Supplementary Fig. 6 and Supplementary Table 1. Error bars represent the SEM, each data point represents the average from three replicas (wells) of an individual patient's CSF. Source data is available as a Source data file.

Uptake of toxic αSyn species by cells and their subsequent intracellular trafficking is a crucial part of the proposed αSyn transmission pathology[20]. Accordingly, we first characterized and assessed the potential of FluoReSyn Reporter-cells to be used as a research tool for investigating the transmission phenomenon. Our results after introducing recombinant αSyn to the culture medium suggested that the Reporter-cells could reliably report the uptake of extracellular αSyn. Similar to other reports[51,53,54], we did not detect direct cytoplasmic internalization of naked monomeric hαSyn by the Reporter-cells (derived from HEK293 cells). We rather observed that the cytosolic uptake required assistance with lipid-based elements as also had been shown in other studies[51,54]. Different from the HEK-based Reporter-cells, we observed that entry of naked recombinant αSyn monomers into primary neurons expressing FluoReSyn is more likely to occur, and lipid-based facilitating agents like the RNAiMAX make the entry process even more efficient. These observations expose the influence of the cellular context on the translocation of αSyn from the extracellular to the intracellular space (e.g., requirement of distinct receptor interactions or membrane translocators for αSyn)[20] and can be interpreted as a reflection of the neuronal nature of αSyn transmission pathology. In line with this interpretation, we did not observe a transmission event when HEK293-based cells stably transfected with hαSyn were co-cultured with the Reporter-cells. On the other hand, these transmission events might be highly dependent on the concentration of hαSyn and duration of exposure. With our setup we could not employ higher concentrations of hαSyn in order to avoid toxic effects to the cells and furthermore, we were limited in terms of exposure durations as a longer maintenance of dividing cells in culture became difficult after several days.

In the current study, we did not observe a positive FluoReSyn signal in transduced neurons following their exposure to fibrillary αSyn. This was not an unexpected observation since fibrils are large structures (in the μm range) that should not enter the cellular cytosol easily, especially as they are known to not form pore-like structures on membranes[55]. Accordingly, it is probable that our observations merely reflected the inability of these bulky structures to penetrate into the neuronal cytosol.

It has been proposed that the uptake of larger arrangements such as oligomers, fibrils, or aggregated αSyn might be mediated by regular endocytosis[20,22]. In this case, FluoReSyn would not detect compartmentalized αSyn in endocytosed vesicles unless αSyn finds a way to escape into the cytosol. It has recently been shown that αSyn pre-formed fibrils (pffs; <50 nm of length[56]) that were internalized by cultured primary neurons remained confined to endo-lysosomal compartments up to 7 days, without a major escape from the endocytic pathway[57]. Thus, even small fibrils have difficulties penetrating into the neuronal cytosol.

Finally, these results do not exclude the possibility that other types of large assemblies or fibrils do penetrate into the cell cytosol, and would therefore be detected by the sensor. This issue could be tested more thoroughly in the future. It is known that many types of fibril-like assemblies with different morphologies and structures can be obtained by changing the preparation and incubation conditions (e.g., pH, salinity, temperature, presence of modulators)[58,59]. Structural variabilities of recombinant fibrils were also shown to propagate to in vitro and in vivo functional properties of the formed assemblies[19,58,59]. For example, it has been shown that upon reducing the incubation pH, the morphology of formed high-molecular weight αSyn assemblies shifted from fibrillary to more amorphous with the latter showing reduced in vitro seeding efficiency[58]. Another study demonstrated that simply by changing from physiological salt concentrations to a lower salt condition, or by adding a chelating agent to the incubation buffer the appearance of the formed high-molecular weight assemblies of hαSyn changed from cylindrical to flat. These assemblies, referred to as fibrils or ribbons, respectively, were shown not only to differ in other structural aspects but also in terms of their in vitro-, in vivo seeding, and propagation properties as well as the degrees of cellular toxicity they induced[59].

Our results demonstrated that the FluoReSyn works both in a cell line and primary neurons, and can report the uptake of recombinant αSyn monomers. Introducing recombinant or tissue derived αSyn extracellularly to in vitro cell cultures is a common approach for investigating the internalization and subsequent seeding activities of αSyn in the context of transmission paradigm[23,60,61]. However, the readily available cellular models and tools are generally limited in terms of offering an unambiguous and spatially resolved discernment of internalized αSyn from the extracellularly applied αSyn seeds[57]. Therefore, there is a necessity to develop novel tools that would enable selective visualization of internalized αSyn and deliver a quantitative characterization of αSyn uptake phenomenon[57]. In a recent study where neuronal cultures were exposed to GFP-tagged αSyn

pre-formed fibrils (pffs), a membrane-impermeable fluorescence quencher dye was used to exclusively quench the fluorescence of extracellular fibrils, thus enabling the selective imaging of only internalized seeds[57]. Similarly, we propose that, FluoReSyn is an optimal tool for this purpose as it can exclusively detect cytosolic αSyn. A major advantage of our system is that it avoids the necessity of employing tagged or covalently conjugated forms of αSyn, which as discussed previously may not recapitulate the normal behavior of native, untagged αSyn. Accordingly, provided that the nanobody epitope is exposed, FluoReSyn can be used for tracking the behavior of any endogenous species of native αSyn obtained from human materials.

Our data suggest that FluoReSyn is a useful research tool not only for neuronal cultures, but also for in vivo set-ups like in our experiments with living *X. Laevis*. Besides, the FluoReSyn's specificity to hαSyn would be of advantage when studying hαSyn transmission in animal models (e.g., mice, rat) as any interference from endogenous αSyn present in rodents can be disregarded.

As a next step in further characterizing the cellular reporter system, we evaluated its ability to detect human-originated αSyn with the future perspective of developing it further into a diagnostic tool[62–65]. Here, we used CSF samples from a readily available cohort of individuals (42). The results showed that our Reporter-cells could indeed detect specific species of αSyn in human CSF, which are able to enter into their cytoplasm. Considering our observations with recombinant αSyn monomers that required a lipid coating agent for entering the Reporter-cells, it is plausible that the αSyn molecules we detected in CSF samples were in vesicular structures such as exosomes as already suggested by others[66,67], or were species other than monomers, such as oligomers with different proposed means of entering the cells[68–71].

The presence of αSyn in CSF can be measured biochemically (e.g., ELISA)[72], and many efforts have been directed at developing assays for detection of CSF-originated αSyn as a biomarker[73,74]. It is important to emphasize that our system provides extra information by detecting hαSyn forms that are able to get into the cytosol of cells. Therefore, it is plausible that we are detecting the species that are more prone to transmission related pathology. This is particularly relevant in the light of emerging evidence, which suggests that particular species of αSyn (e.g., oligomers, aggregates) that are associated with toxicity can serve as far-better biomarkers than total αSyn in CSF[74].

Although our initial results in Fig. 1 show that the FluoReSyn signal correlates to the amount of αSyn, our Reporter-cells are not yet quantitative enough to precisely determine the concentrations of transmittable αSyn in human CSF. Nevertheless, this is a fascinating first proof-of-concept. To the best of our knowledge, our cellular hαSyn reporter system is so far the sole approach developed for the detection of transmittable hαSyn species present in human body fluids. We believe that major optimizations can be performed to this system to increase its sensitivity and accuracy. These enhancements might include employment of a faster maturing and brighter EGFP variant, tuning of the FluoReSyn sensitivity biochemically, and destabilizing the antigen-unbound NbSyn87 (ref. [75]). Once these optimizations are performed and validated, a logical next step would be the generation of a knock-in (KI) mice, to thus generate an animal model for studying the mechanical and molecular aspects of αSyn transmission. Using primary cells from these KI mice, it will then be possible to establish a neuron-based reporter system, which would presumably hold more promise in a diagnostic context. We hope such efforts might help to achieve more sensitive and quantitative read-outs, which would in turn pave the way for the development of an accurate and reliable diagnostic or prognostic tool for αSyn associated disorders.

## Methods

**Plasmid transfections and virus infections.** Transfections with pcDNA 3.1 (+) vectors (Thermo Fisher Scientific) encoding for hαSyn (accession number NM_000345.4), human β-synuclein (hβSyn; accession number NM_001001502.3) or rat αSyn (rαSyn; accession number NM_019169.2) sequences were performed with Lipofectamine 2000 (Invitrogen, Thermo Fisher Scientific) and Opti-MEM I Reduced Serum Medium (Opti-MEM from Gibco, Thermo Fisher Scientific, Waltham, MA, USA) according to manufacturer's instructions. Transfected cells were typically used after 48 h. The AAV coding for NbSyn87 fused to mCherry and nuclear localization signal (NLS) sequences (NbSyn87-mCherry-NLS) was kindly provided by Dr. Sebastian Kügler, Department of Neurology; Viral Vectors Laboratory, University Medical Center Göttingen, Germany.

**Protein purification.** Rpn10, hαSyn, hβSyn were produced using NEB Express Competent *Escherichia coli* (New England BioLabs Inc., Ipswich, MA, USA). Bacteria were grown overnight with the plasmid of interest in Lysogeny Broth (Sigma-Aldrich) and the respective antibiotics. Next day, cells were further cultured in Terrific Broth (Sigma-Aldrich) and antibiotics until OD was ~2–3 at 37 °C and induced with 0.8 mM IPTG for 4 h. After adding 5 mM EDTA, cells were harvested by centrifugation at ~ 3000 × g for 30 min at 10 °C and frozen at −20 °C until further processing. Pellets were resuspended on ice with 1 mM DTT, 25 mM Imidazole, and 1 mM PMSF in a His-binding buffer (50 mM HEPES pH 8.0, 500 mM NaCl, 5 mM MgCl$_2$ and 10% glycerol) and bacteria were lysed by sonication on ice. Cell debris was separated by centrifugation at ~ 15,000 × g for 1 h at 4 °C. Clear supernatant was incubated with 2 ml of pre-equilibrated Ni-beads (cOmplete His-Tag Purification Resin, Roche, Switzerland) for 1 h and then transferred to a sigma column to be washed consecutively with His-binding buffer (50 mM HEPES pH 8.0, 1000 mM NaCl, 10 mM MgCl$_2$, 25 mM Imidazole and 5% glycerol) and low-Salt-pH buffer (50 mM HEPES pH 7.5, 500 mM NaCl, 5 mM MgCl$_2$, 25 mM Imidazole, 5% glycerol) with each of them containing 25 mM Imidazole. Finally, protein was eluted by 500 mM Imidazole and His-Tag was cleaved with SUMO protease and removed by reverse binding to nickel beads. Pure proteins were desalted into PBS, purity was confirmed by polyacrylamide gels (PAGE) and concentrations were determined using nanodrop spectrophotometer considering the protein molecular weight and extinction coefficient.

**NbSyn87 coupling to Alexa647 fluorophore.** NbSyn87 protein was obtained from a custom production service offered by NanoTag Biotechnologies GmbH (Göttingen, Germany), and the nanobody was equipped with one ectopic cysteine at its C-terminal. 1 mg of pure NbSyn87 was reduced by adding TCEP (Sigma-Aldrich) to a final concentration of 5 mM for 1 h. The reduced sample was then desalted using gravity columns Nap10 (GE Healthcare Life Sciences) in nitrogen bubbled PBS (pH 7.4) and immediately added to five molar excess of maleimide-functionalized Alexa647 (Thermo Scientific) for 1 h. Excess of free dye was separated from the conjugated nanobody with an Äkta HPLC equipped with a Superdex 75 increase column (GE Healthcare Life Sciences).

**Generation of hαSyn fibrils.** Monomeric hαSyn was expressed recombinantly in *E. coli* BL21 (DE3) and purified by anion exchange and size-exclusion chromatography[58]. Cell lysis was conducted by French press (Avestin EmulsiFlex-C3) with lysis buffer (10 mM Tris-HCl, pH 8, 1 mM EDTA, 1 mM PMSF), 20 mL per 1 L of cell culture. The lysate was then heated up to 96 °C for 30 min in a water bath and centrifuged afterwards for 30 min at 4 °C with 22,000 × g (Beckman Coulter, JA 25-5 rotor). Streptomycin was added at a final concentration of 10 mg/mL and incubated for 15 min. Following another centrifugation step, the soluble protein was precipitated by ammonium sulfate for 15 min at 4 °C. Protein pellet was dialyzed overnight against 25 mM Tris-HCl, pH 7.7 and loaded on an anion exchange column (GE Healthcare, Mono Q 5/50 GL). Protein was eluted with 300 mM NaCl. Monomeric hαSyn purity and was achieved by size-exclusion chromatography (GE Healthcare, Superdex 75 10/300 GL) using 50 mM HEPES, pH 7.4, 100 mM NaCl, and 0.02% NaN$_3$. The protein was sterile filtered (0.22-μm) and stored at 1 mM at −80 °C. Monomeric hαSyn in 50 mM HEPES, pH 7.4, 100 mM NaCl, and 0.02% NaN$_3$ was centrifuged at 84,000 × g for 1 h at 4 °C. The supernatant was filtrated through 0.22 μm ULTRAFREE-MC centrifugal filter units (Millipore) and adjusted to 0.25 mM protein concentration. Aggregation was performed for 10 days at 37 °C with constant stirring at 200 rpm. Progress of fibril formation was monitored with a Thioflavin T fluorescence assay[76]. The fibrils were finally collected by ultracentrifugation at 20 °C, washed twice with 50 mM HEPES, pH 7.4, 100 mM NaCl and quantified by subtracting the amount of monomeric hαSyn in the supernatant from the total protein used for aggregation. Prior to their use, fibrils were resuspended in the washing buffer at 0.1 mM protein concentration.

**Proteasome inhibition.** MG132 (Sigma-Aldrich, St. Louis, MO, USA) was administrated at different concentrations (Supplementary Fig. 2) or at 1 μM to the culture medium for different time intervals (Fig. 2).

**CSF samples.** Study participants consisted of individuals who were in treatment at the Paracelsus Elena Klinik, Kassel, Germany, and had been diagnosed with a

variety of neurological disorders non-related to αSyn aggregation disorders. The study cohort consisted of 23 females and 19 male individuals with a mean age of 70.95 ± 1.51. For a detailed presentation of demographic and clinical features of participants please see Supplementary Table 1. CSF samples from all individuals were collected after the informed consent of the participant at the Paracelsus Elena Klinik in accordance with the principles of Declaration of Helsinki and following identical standard operating procedures. In brief, CSF was collected by lumbar puncture in the morning with fasting patients in a sitting position. The samples were centrifuged at $2000 \times g$ for 10 min at room temperature (RT). The concentrations of αSyn in CSF samples were measured with a validated sandwich ELISA system (mSA1/Syn1-BB; 384 well plate format)[72]. Aliquots of the supernatants were frozen within 20–30 min and stored at −80 °C until their use. A total cell count was established in tube 1 (2 mL). Samples with erythrocyte counts >50 cells per µL CSF in tube 1 were excluded from all analyses. The use of the CSF samples in this study was approved by the ethical committee of the Medical Center Göttingen with the approval numbers 36/7/02 and 9/7/04.

**Cell lines.** Wild-type HEK293 were grown in Dulbecco's modified Eagle's medium (DMEM) supplemented with 10% FBS, 4 mM L-glutamine and 600 U/ml penicillin-streptomycin (Lonza). Reporter-cells (HEK293 stably expressing TetON-NbSyn87-EGFP-P2A-mCherry) and hαSyn cells (HEK293 stably expressing TetON-hαSyn) were grown in DMEM supplemented with 10% FBS, 2 mM L-glutamine supplemented with 0.5 µg/ml puromycin (InvivoGen, San Diego, CA, USA). Both, Reporter-cells and hαSyn cells were produced by Sirion Biotech GmbH (Martinsried, Germany). TetON Induction was performed using 0.5 µg/ml doxycycline (Sigma-Aldrich, St. Louis, MO, USA) at least 12 h before their use.

**Primary hippocampal neurons.** Postnatal (P1–P2) pups from Wistar rats were decapitated and the brains were extracted. The hippocampi were isolated, washed in Hank's balanced salt solution (HBSS; Invitrogen, Darmstadt, Germany) and incubated in enzymatic digestion solution for 1 h at room temperature. After washing in HBSS the hippocampi were incubated in inactivation solution for 15 min. After another washing step in Neurobasal A, neurons were mechanically dissociated by pipetting. In all, 15,000 neurons per well were added to the plating medium (MEM, 10% horse serum, 3.3 mM glucose, 2 mM glutamine) in PLL-coated 96-multiple glass bottom well plates (SensoPlate, Greiner Bio-One International GmbH, Kremsmünster, Austria) and kept at 37 °C, 5% CO₂. After ~ 1 h, when the neurons adhered to the glass bottom, the plating medium was exchanged with 100 µL Neurobasal A and plates were further cultured at 37 °C, 5% CO₂. To maintain healthy cultures, 50 µL of medium was removed every second day and replaced with 50 µL of fresh Neurobasal A. The primary rat hippocampal neuron cultures were prepared with minor modifications from the original protocol[77]. Five days after plating, neurons were infected with an AAV (88e5 TU AAV/5 µl) containing the sequence for NbSyn87 fused to mCherry and NLS sequences (NbSyn87-mCherry-NLS).

**Cellular uptake of recombinant hαSyn from the culture medium.** Purified synuclein proteins were introduced to the medium of (i) neurons cultured on 96-well plates 6 days post infection with AAV NbSyn87-mCherry-NLS and (ii) Reporter-cells cultured on 24-well plates 48 h (~70,000 cells per well) post-seeding. hαSyn (monomers or fibrils) and hβSyn proteins were diluted in Opti-MEM and incubated with Lipofectamine RNAimax (Invitrogen, Thermo Fisher Scientific, 2 and 0.3 µl per well for Reporter-cells and neurons, respectively) for 20 min and then administered to the culture medium at a final concentration of 20 µM protein per well. Cells were fixed 14 h later for imaging and evaluation.

**Detection of hαSyn species in CSF samples.** Wild-type HEK293 and Reporter-cells were seeded (~ 14,000 per well) and induced in a PLL-coated 96-well plate with glass bottom (SensoPlate, Greiner Bio-One International GmbH, Kremsmünster, Austria). After 24 h of induction, the cells were exposed to CSF samples. In brief, the medium was partially eliminated leaving 25 µl per well, and supplemented with 50 µl of CSF or culture medium as negative control. Cells were further incubated for 24 h and then fixed by adding 4% paraformaldehyde in PBS (137 mM NaCl, 2.7 mM KCl, 10 mM Na₂HPO₄, 2 mM KH₂PO₄; pH 7.4) overnight at 4 °C, aldehyde groups were quenched with 0.1 M NH₄Cl for 15 min and images were acquired.

**Immunostaining.** Cells were briefly washed with DPBS (Reporter-cells) or Tyrode buffer (primary rat hippocampal neurons), fixed with 4% paraformaldehyde in PBS for 30 min at RT. Remaining reactive aldehyde groups were quenched in PBS supplemented with 0.1 M glycine and 0.1 M NH₄Cl for 15 min at RT. Cells were permeabilized and unspecific protein binding sites were blocked with a blocking/permeabilization solution (0.1% Triton X-100 and 2% bovine serum albumin, BSA in PBS) at RT for 15 min. Reporter-cells were incubated with a rabbit polyclonal anti-α/β-Synuclein antibody (dilution 1:500; Cat. No. 128002, SySy, Göttingen, Germany) and neurons with a polyclonal guinea pig anti-synaptophysin antibody (dilution 1:100 Cat. No. 101004, SySy, Göttingen, Germany) for 1 h. After three thorough washing steps, cells were incubated with the secondary antibodies (donkey anti-rabbit, dilution 1:500; Cat. No. 711-175-152, Dianova, Hamburg, Germany or donkey anti-guinea pig IgG labeled with ATTO 647 N, dilution 1:500; Cat. No. N0602-At647N-S, SySy, Göttingen, Germany) for 45 min or 1 h,

respectively. The samples were again subjected to three thorough washing steps with PBS and high-salt PBS (500 mM NaCl, 2.7 mM KCl, 10 mM Na₂PO₄, 2 mM KH₂PO₄; pH 7.3–7.4). Before mounting the coverslips or before imaging, cells were stained with Höchst 33342 (1 µg/ml; Thermo Fisher Scientific). Coverslips were finally mounted in Mowiol mounting media (6 g glycerol, 6 ml deionized water, 12 ml 0.2 M Tris buffer pH 8.5, 2.4 g Mowiol 4-88 from Merck). In all, 96-well plates were imaged in PBS.

**Plasmid electroporation of *Xenopus laevis* tadpoles.** All procedures for animal handling were approved by the governmental animal care and use office (Niedersächsisches Landesamt für Verbraucherschutz und Lebensmittelsicherheit, Oldenburg, Germany, Az.12/0779) and were in accordance with the German Animal Welfare Act as well as with the guidelines of the Göttingen University Committee for Ethics in Animal Experimentation.

*X. laevis* tadpoles (albinos, stage 53)[78] were used for the in vivo experiments. Injection micropipettes were pulled from borosilicate microcapillaries (Warner instruments; outer diameter: 1.0 mm, inner diameter: 0.58 mm, length 100 mm) using a horizontal puller (P 1000, Sutter Instruments). Micropipette tips were sharpened at an angle of 20–30° until the pipette tip had a syringe-like shape (Micropipette Beveler 48000; World Precision Instruments). Micropipettes were filled with 3 µl of plasmid solution/s (600 ng/µl). Cascade blue dextran (3 mM, 10%, Thermo Fisher) was added to the plasmid solution before to be able to observe dye extrusion under fluorescent illumination. Albino tadpoles were anaesthetized in 0.02% MS-222 (ethyl 3-aminobenzoate methanesulfonate; Sigma-Aldrich; pH: 7.6) for 5 min until complete immobility and irresponsiveness. Subsequently, the animal was transferred to a moistened dish under a stereomicroscope with brightfield and fluorescent illumination (Olympus SZX16; light source: X-Cite Series 120 Q, Lumen dynamics). The glass pipettes filled with plasmid solution/s were mounted to a micromanipulator connected to a FemtoJet injection system (Eppendorf). The micropipette was carefully penetrated into the olfactory mucosa at three to five different locations without injuring major arteries. Up to five pressure pulses of 250–1000 hPa (1 s each) were applied per site. Once homogenous blue fluorescent signal was visible throughout the olfactory mucosa, an external electric field was applied to the olfactory mucosa using an electroporation setup[79]. One of the 0.2 mm platinum wire electrodes was positioned in the water-filled nostril, the other one in contact to the skin, laterally to the olfactory nerve. Trains of three square pulses (20 V, 500 ms duration, and 25 ms delay) were applied four times in alternating polarity (ELP-01D, NPI Electronics; additional capacitor connected in parallel: Domoport, 3 µF). The entire procedure was performed in >5 min to grant proper anesthesia and avoid dissipation of plasmid solution before electroporation. Following electroporation, animals were put into water until they woke from anesthesia. After assessment of normal swimming the larvae were left in their aquaria to recover for at least 24 h. Individual nostrils were electroporated sequentially, with a 1 h recovery period in between. We injected and electroporated two plasmids in the left nostril: one expressing FluoReSyn and the other hαSyn, both under the CMV promotor. In the right nostril, only the FluoReSyn expressing plasmid was electroporated.

**Imaging.** Conventional epifluorescence images of the Reporter-cells were obtained with an Olympus IX71 microscope equipped with a 0.5 NA/×20 dry UPlanFL N objective and captured with an Olympus F-View II CCD camera (Olympus, Hamburg, Germany). Experiments with CSF samples and neurons on 96-well plates were acquired using a Biotek Cytation 3 Imaging Reader (BioTek Instruments, Winooski, VT, USA) equipped with a ×20, Plan Fluorite WD 6.6 NA 0.45 objective, a 465 nm LED cube (Cat# 1225001), EGFP Filter cube (Cat# 1225101), 523 nm LED cube (Cat# 1225003), RFP filter cube (Cat# 1225103) and a 16-bit monochromatic CCD camera (pixel size 6.45 µm × 6.45 µm).

**In vivo multiphoton imaging of the *Xenopus* olfactory system.** For in vivo imaging, we anesthetized the electroporated tadpoles in 0.02% MS-222 (ethyl 3-aminobenzoate methanesulfonate; Sigma-Aldrich; pH: 7.6) for 5 min until complete irresponsiveness. The animals were placed into a recess of a silicone-filled recording chamber. The chamber was filled with water and the animal was mechanically fixed using parafilm. A small window was cut into the parafilm to expose the olfactory organs. Both nostrils were imaged under a two-photon microscope (Nikon A1R MP) at an excitation wavelength of 920 nm. All animals ($n = 9$) and nostrils were imaged as 3D image stacks under the same gain and laser settings, to compare fluorescence intensities of the FluoReSyn. The imaging procedure did not last longer than 10 min and animals were transferred to a big water-filled beaker until they recovered from anesthesia.

**Cell lysate preparation.** Reporter-cells were washed briefly with ice-cold DPBS and lysed with 50 µl of lysis buffer per well (50 mM Tris/HCl pH 7.5, 150 mM NaCl, 2 mM EDTA, 0.5% IgePAL, 0.5% Sodium deoxycholate and freshly added 250 µM PMSF, 10 ng Leupeptin, 10 ng Aprotinin, 1 ng Pepstatin A, 10 ng DNase und 1 µl Halt Protease Inhibitor Cocktail; Thermo Fisher Scientific). Cell lysates were collected into a pre-cooled tube and were centrifuged for at least 1 h at ~15,000 × g at 4 °C. The supernatant was collected into two tubes, snap-frozen with liquid nitrogen and stored at −80 °C until needed.

**Western blotting**. Reporter-cell lysates were thawed on ice and diluted accordingly to their total protein (determined using BCA assay (Merk)) to load the same total protein concentration in each lane. Samples were mixed with pre-heated 5× Laemmli buffer (50 mM Tris-HCl, 4% sodium doedecyl sulfate (SDS), 0.01% Serva Blue G, 12% glycerol, pH 6.8, 50 mM DTT) to be further boiled at 95 °C for 10 min, centrifuged and then loaded into previously casted 10–12% PAGE. After the SDS-PAGE run was completed, proteins were transferred to a nitrocellulose membrane in wet trans blot cell (Biorad) with 400 mA for 2 h at 4 °C while stirring the transfer buffer (25 mM Tris, 192 mM glycine, pH 8.3 and 20% methanol, and 0.04% SDS). The membrane was incubated for 1 h in blocking buffer (5% Nonfat Dried Milk, 0.1% Tween20 in PBS) and then was further incubated with a mouse monoclonal anti-EGFP antibody (1:500; Cat. No. A11120, Invitrogen, Thermo Fisher Scientific) or a rabbit polyclonal anti-β-Actin-Cy5 antibody (1:1000; Cat. No. 251003, SySy, Göttingen, Germany). Primary antibody incubations were performed overnight at 4 °C with constant shaking. The following day, the membrane was washed thoroughly in blocking buffer and incubated with the fluorescently labeled secondary donkey polyclonal anti-mouse antibody (1:1000; Cat. No. 715-175-150, Dianova) for 1 h at RT. For experiments with the directly labeled anti-β-Actin antibody, the second 1 h incubation step was omitted. Membrane was washed several times each in 0.1% Tween20 in PBS and imaged using an Amersham Imager 600 (GE Healthcare Life Sciences, Little Chalfont, UK).

**Dot-Blot**. Proteins were serial diluted in washing buffer (0.05% Tween20 in PBS) and spotted on a nitrocellulose membrane. After the membrane was dried, it was blocked with 2% FBS or BSA, 5% Nonfat Dried Milk, 0.05% Tween20 in PBS for 1 h under agitation. This was followed by incubation with the fluorescently labeled nanobody NbSyn87-Alexa647 for 1 h. Unbound nanobodies were washed away by several thorough washing steps with 0.05% Tween20 in PBS for a total duration of 1 h. Finally, images of the membranes were taken with an Amersham Imager 600 (GE Healthcare Life Sciences, Little Chalfont, UK) to detect the NbSyn87-Alexa647 signal.

**Data analysis and statistics**. Image analyses of experiments presented in Figs. 1, 2, 4–7, and Supplementary Figures were performed with custom-written procedures in Matlab (MathWorks Inc., Natick, MA, USA). Cells were identified automatically based on their Höchst 33342 (Figs. 1, 2, 4–5, Supplementary Fig.) or mCherry (Fig. 7) signals. The average signal intensity within a cell was calculated and corrected for the background intensity by subtracting the background region of interest from the average signal intensity.

Cells were considered as positive if their background-corrected GFP (Figs. 1, 4, 5, Supplementary Fig. 3, 4) or mCherry (Fig. 6) signal intensity (AU) was above the mean plus 2 standard deviations of control GFP or mCherry signal intensity, respectively. For Fig. 7, cells were considered as positive if background-corrected GFP signal intensity was above 400 AU. In Fig. 7, signal intensity was normalized to mCherry to exclude differences in FluoReSyn induction.

Graph plotting as well as statistical analyses of data presented in Figs. 1, 2, 4–7, and Supplementary Figures were carried out using custom-written procedures in Matlab, Sigma Plot (Systat Software, San Jose, CA, USA) or GraphPad Prism 5.0 (San Diego, CA, USA). All values are given as mean ± standard error of the mean from at least three independent experiments. Statistical significance was assessed by one-way ANOVA and Tukey's Post hoc test.

Images showing FluoReSyn signal in vivo (Fig. 3) were created from multi-channel 3D image stacks acquired with the multiphoton microscope. Autofluorescence from melanophores was removed by subtracting the maximum intensity z-projections of the blue emission channel (where only autofluorescence was visible) from the green emission channel (with FluoReSyn signal only) using Fiji "Image Calculator" function[80]. For quantitative analysis of differences in FluoReSyn fluorescence between nostrils, maximum intensity z-projections of the green emission channel were processed in Fiji to obtain a binary mask of the regions of interest (nuclei with FluoReSyn signal). Therefore, binary images were created using Li's method of minimum cross-entropy thresholding[81] followed by four rounds of despeckling in Fiji[80]. The resulting binary masks were used to measure mean fluorescence intensity values of all regions of interest in the images with areas between 20 and 150 px (size range of nuclei). Fluorescence mean intensities were pooled for each condition (FluorReSyn + hαSyn vs. only FluoReSyn) from nine (559 nuclei in total) animals measured under the same conditions. Scatter plots were created using the Seaborn package in Python (python.org; version 0.9; 10.5281/zenodo.1313201).

**Reporting summary**. Further information on research design is available in the Nature Research Reporting Summary linked to this article.

## Data availability
The data that support the finding of this study is readily available within this paper, its Supplementary file and in the Source Data file. The data set for Figs. 1e, 2b, c, 3i, 4b, 5b, 6b, c, 7b and Supplementary Fig. 1a–c, 3a, b, 5b, and 6 are provided in the Source Data file. All data sets generated during and/or analyzed during the current study are available from the corresponding author on reasonable request.

## Code availability
The customized code generated during and/or analyzed during the current study is available from the corresponding author on reasonable request.

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

## Acknowledgements

F.O., F.B.B., I.M., and T.O. were supported by the *Deutsche Forschungsgemeinschaft* (DFG) through Cluster of Excellence Nanoscale Microscopy and Molecular Physiology of the Brain (CNMPB). We thank Dr. Sebastian Kügler for providing us the AAVs and helpful discussions. We thank Karin Giller, Melanie Wegstroth, Christina Schäfer, and Nicole Hartelt for excellent technical help. Supported by the DFG under Germany's Excellence Strategy - EXC 2067/1- 390729940. Supported in part by the DFG through grant SFB1286/Z3 to S.O.R.

## Author contributions

C.G., T.O., and E.F.F. designed and conducted experiments, analyzed and interpreted data, and contributed with the writing of the manuscript. N. Waal and H.V. designed and performed experiments, analyzed and interpreted data. N. Wender initially characterized the sensor, designed the Reporter-cell lines, designed and performed experiments, analyzed and interpreted data. I.M. contributed to the interpretation of data and scientific discussions. C.T. and B.M. provided the human CSF samples. T.S., M.Z., and S.B. prepared and provided αSyn fibrils. S.O.R. analyzed and interpreted the data, contributed to the supervision of the study and scientific discussions. F.B.B. designed and performed experiments, analyzed and interpreted the data, supervised the study and wrote the manuscript. F.O. conceived the project, designed and performed experiments, analyzed and interpreted the data, supervised the study and wrote the manuscript.

## Competing interests

F.O. and S.O.R. are shareholders of NanoTag Biotechnologies GmbH. All other authors declare no competing interests.
