## [Peer Review File · Nature Communications]

Reviewers' comments:

Reviewer #1 (Remarks to the Author):

The authors show that the nanobody with specificity towards α -synuclein has a cross reactivity to an amino acid stretch that also occurs in Rpn10, a protein at the entry of the proteasome. As a consequence this nanobody tagged with fluorescent protein (FP) is only short lived in cells (rapidly degraded by the proteasome) unless α -syn is also present in these cells (or if proteasomal activity is blocked with MG132).

This finding is exploited to generate a sensor for scoring the intracellular presence of human α -syn either produced inside the cells or transmitted inside the cell, by various pathways. The sensor (reporter cell) is specific for the α -syn (rat or human) as the epitope seems to be conserved as well as the conserved epitope of Rpn10 that cross reacts with this nanobody. These reporter cells also detect any transmittable form of the human α -syn, such as that in human CSF. A minor criticism is at its place: the correlation among the amount of α -syn in the CSF and the signal in the reporter cells is not obvious from the manuscript.

Furthermore, although the findings described in these report are of general interest, it is not clear whether this reporter or screening system could be broadened to other targets beside α -syn (by making use of bispecific nanobody constructs). Probably other monomeric nanobodies with other epitope specificity will not work as they will not cross react to the rpn10 or any other proteasome epitope.

Finally, a discussion is missing on the general sensitivity of intracellular nanobodies to degradation if their target is absent. It was indeed shown by Keller BM (group of Rothbauer U; papers in Mol Cell Proteomics and Antibodies) that intracellular nanobodies are rapidly degraded in the cytosol (probably independent on targeting the Rpn10 peptide).

Reviewer #2 (Remarks to the Author):

Gerdes et al describe a nanobody-based reporter that they have developed for the detection of monomeric forms of alpha-synuclein. They show that NbSyn87 may be stabilized by binding to human aSyn, preventing its degradation in the proteasome, to which it is otherwise recruited by weak binding to Rpn10.

I have two major comments, one related to the use cases for this reporter, and one related to the experiments reported in this paper.

1. As shown in Figure 6, FluoReSyn detects primarily monomeric forms of human aSyn (no signal with introduction of haSyn fibrils). It is not clear exactly what the use case of such a reporter might be – in most disease contexts, it is the detection of pathological forms of haSyn that is most valuable.

2. With respect to the experiments reported here, the only context in which any type of dose curve is shown is in Figure 1, where variable amounts of haSyn plasmid are transfected into HEK293 cells, and higher levels appear to result in a higher % of positive reporter cells. However, as shown in Figure 6c, variability under the same conditions is pretty large – replicates treated with haSyn and iMAX show anywhere from 10% to 35% FluoReSyn+ neurons. This assay is really only useful if there is some ability to quantify aSyn, which is minimally shown by the authors (with some data suggesting that in most use contexts, variability is too high for meaningful quantification).

Reviewer #3 (Remarks to the Author):

This is a very interesting paper, with a significant number of strong experiments to support the conclusion that a fluorescent fusion of the NbSyn87 nanobody can serve as a reporter for the presence of human α -Syn in cellular cytoplasm. The range of assays used to measure both target that is over-produced in transfections, as well as penetration of exogenously introduced protein, form a convincing series. Methods descriptions seem very detailed.

The authors have acknowledged in the discussion that there are multiple variables of cell lines, culture conditions, and protocols used to produce α -Syn for the extracellular experiments. It might be worthwhile to explore these questions in a bit more detail, since they are very critical to the interpretation of the results. Not all readers will have the time or access to look up the cited papers. The 7-line final sentence makes it clear that the authors plan to do major optimizations, presumably including much more neuronal cell lines as more faithful reporters. Their more explicit expert opinion on which directions to prioritize could be a useful addition to the field.

The only experiment that seems to have been incompletely reported is the one testing the human CSF from a range of disorders. While it is clear that the HEK293-derived reporter line is more proof of general concept than definitive, it would still be useful to see the actual distribution of the different diseases in this assay. Since the data are presented individually, it would seem possible to do this, in which case the scatter plot would become a regular, rather than a supplemental, figure.

Reply to the reviewer comments

Reviewer #1:

The authors show that the nanobody with specificity towards α -synuclein has a cross reactivity to an amino acid stretch that also occurs in Rpn10, a protein at the entry of the proteasome. As a consequence, this nanobody tagged with fluorescent protein (FP) is only short lived in cells (rapidly degraded by the proteasome) unless α -syn is also present in these cells (or if proteasomal activity is blocked with MG132).

This finding is exploited to generate a sensor for scoring the intracellular presence of human α -syn either produced inside the cells or transmitted inside the cell, by various pathways. The sensor (reporter cell) is specific for the α -syn (rat or human) as the epitope seems to be conserved as well as the conserved epitope of Rpn10 that cross reacts with this nanobody. These reporter cells also detect any transmittable form of the human α -syn, such as that in human CSF.

We thank the reviewer for these comments. To avoid misunderstandings, we would like to clarify that our sensor is specific for human α Syn. The epitope is not perfectly conserved between human, rat or mouse (as shown in Supp. Fig. 3). The sensor is therefore unable to detect rat α Syn (Figure 4), and only responds to human α Syn.

A minor criticism is at its place: the correlation among the amount of α -syn in the CSF and the signal in the reporter cells is not obvious from the manuscript.

We thank the reviewer for this suggestion. We have now added a series of graphs (the new Supp. Fig. 6) displaying the Reporter-cell signal in relation to the clinical report of the patients, as well as correlating it with the α Syn concentrations in CSF samples. We want to remark here the point that the sensor will only detect h α Syn that enters into the cytoplasm of a Reporter cell, thus it is not necessarily correlating to total α Syn present in the patient CSF.

Due to the limited number of patient samples we had access to, with different pathologies, we were unable to find conclusive correlations between the clinical phenotypes and the Reporter-cell responses. Future studies, with larger patient cohorts, should test whether the FluoReSyn system can be used as a diagnostic tool in a large scale.

Along these lines we also modified the text in Figure 7 and the Discussion of the manuscript to make this clearer.

Furthermore, although the findings described in these report are of general interest, it is not clear whether this reporter or screening system could be broadened to other targets beside α -syn (by making use of bispecific nanobody constructs). Probably other monomeric nanobodies with other epitope specificity will not work as they will not cross react to the rpn10 or any other proteasome epitope.

The reviewer is right that other monomeric nanobodies would not work in this fashion unless they also by chance have an affinity towards Rpn10. We also agree with the reviewer that it will be of significant interest to be able to apply this tool to a large number of other targets. The NbSyn87 nanobody (used for FluoReSyn) could be used in a general manner by introducing the human α Syn epitope on the protein of interest. This strategy could work as an example by introducing the α Syn epitope at the genomic level with CRISPR/Cas9 to tag endogenous proteins of interest. By doing this, the sensor could be used for nearly every

protein with the only disadvantage of having altered the endogenous protein by adding ~12 amino acids to it. Obviously, this can only be envisioned in cells where no human α Syn is endogenously present, to avoid cross-reactivity.

Following on the lead of the reviewer, a similar strategy could be developed by using a bispecific nanobody construct where NbSyn87 nanobody would be combined with a second nanobody that is specific for another protein. In principle, the binding of NbSyn87 to the conserved epitope of Rpn10 would direct the whole construct to degradation, unless the second nanobody finds its target, and brings the whole construct to its cellular location, thus preventing degradation. However, depending on the concentrations of all players involved, and on the affinity of the second nanobody, it is possible that the bispecific construct actually shuttles the target protein to degradation, which would prevent this construct from working as a general sensor system. Such constructs would therefore need to be carefully designed, tested and optimized.

Finally, a discussion is missing on the general sensitivity of intracellular nanobodies to degradation if their target is absent. It was indeed shown by Keller BM (group of Rothbauer U; papers in Mol Cell Proteomics and Antibodies) that intracellular nanobodies are rapidly degraded in the cytosol (probably independent on targeting the Rpn10 peptide).

This is an important point that has been difficult to tackle in the field of nanobodies/intrabodies. The affinities and thermostabilities of nanobodies are always measured *in vitro*, and we are not aware that it has been possible to measure them inside living cells, so in practice it is difficult to estimate the fraction of the nanobodies that are bound to their targets in the cytosol. The main problem here is that some nanobodies are intrinsically unstable in the cytosol, and tend to be incorrectly folded, which renders them impractical for use as intrabodies. Binding to the target may stabilize the nanobodies, but not in all cases, as some nanobodies will misfold irrespective of the target presence. In such cases the nanobodies will tend to form large aggregates in the perinuclear area, or will be removed by the proteasome system. The opposite example comes from our recently published nanobody anti-ALFA-tag (PMID: 31562305). This nanobody works as an intrabody, and we do not observe that it is being degraded rapidly in the cytosol.

Until recently, there were no general guidelines to predict which nanobodies would (or wouldn't) work as an intrabody. One hint is the presence of 2 or more disulfide bonds, since nanobodies containing several disulfide bonds have low chances of working as intrabodies. A very interesting study from Tang et al. (PDB: 27205882) shows that by mutating some specific amino acids on nanobodies it is possible to destabilize them in the cell cytosol. Tang and colleagues used the anti-GFP nanobody from Rothbauer (which is stable in the cytosol), and they found 6 amino acid positions that can destabilize nanobodies. Three are mild destabilizers, and 3 are major destabilizers. The SynNb87 nanobody used here has indeed only one of the minor destabilization mutations. All other 5 positions contain amino acids considered as providing stability to the nanobody scaffold, which suggests that this nanobody would normally be stable in the cytosol.

Reviewer #2:

Gerdes et al describe a nanobody-based reporter that they have developed for the detection of monomeric forms of alpha-synuclein. They show that NbSyn87 may be stabilized by binding to human α Syn, preventing its degradation in the proteasome, to which it is otherwise recruited by weak binding to Rpn10.

I have two major comments, one related to the use cases for this reporter, and one related to the experiments reported in this paper.

We thank the reviewer for the comments.

1. As shown in Figure 6, FluoReSyn detects primarily monomeric forms of human α Syn (no signal with introduction of h α Syn fibrils). It is not clear exactly what the use case of such a reporter might be – in most disease contexts, it is the detection of pathological forms of h α Syn that is most valuable.

We apologize for this confusion. We recognized after the reviewer's comment that the way we had described the fibril experiments in the initial manuscript was not optimal and was therefore prone to misunderstandings.

The nanobody on which FluoReSyn was derived is fully able to detect fibrillar forms of α Syn. The nanobody has been characterized thoroughly *in vitro* by the group of Christopher Dobson (PMID: 23557833). They demonstrated that the nanobody can bind equally well to monomeric α Syn and to fibrils. We have also replicated this finding, by showing that the nanobody was capable of binding to our *in vitro*-generated fibrils (Fig. 6d).

Fibrils from the same batch used for *in vitro* detection were employed later by adding them to the culture medium of neurons. There the fibrils were not detected by the sensor (Fig. 6). This is not due to the inability of the sensor to bind them, but rather only an indication that such fibrils did not penetrate into the cells, and therefore could not be detected. α Syn fibrils are objects of several micrometers (see for example PMID: 23557833), and therefore we do not expect to find such structures entering the cytosol of living cells. We have now clearly addressed this argumentation in the Results and Discussion sections of the revised manuscript. We have also modified the wording of other discussion points about the interpretations and implications of experiments with fibrillary α Syn for enhancing clarity.

Our observations are in agreement with several theories that advocate for oligomers and/or pre-formed fibrils (pffs) as the toxic species, but typically not full fibrils (PMID:19888725). A general hypothesis is that oligomers or α Syn mutants (PDMI: 19745811) might be able to enter the cytosol and act as nucleator elements, triggering the formation of oligomers or fibrils that eventually cause cellular dysfunction. In spite of current debates in the field, few works indicate that large fibrils might penetrate from the outside into the cell cytoplasm. We therefore see these results as simply in line with current hypotheses on α Syn behavior: the sensor could not respond to the fibrils, because the fibrils could not penetrate into the cytosol. Even small pre-formed fibrils have major difficulties in entering the cell cytosol (PMID: 28611062).

We have now explained the experiments and their results in more detail, to increase clarity on this matter.

2. With respect to the experiments reported here, the only context in which any type of dose curve is shown is in Figure 1, where variable amounts of h α Syn plasmid are transfected into HEK293 cells, and higher levels appear to result in a higher % of positive reporter cells. However, as shown in Figure 6c, variability under the same conditions is pretty large – replicates treated with h α Syn and iMAX show anywhere from 10% to 35% FluoReSyn+ neurons. This assay is really only useful if there is some ability to quantify α Syn, which is minimally shown by the authors (with some data suggesting that in most use contexts, variability is too high for meaningful quantification).

The first figure of the manuscript was performed to characterize FluoReSyn. For this, we used Reporter-cells that were expressing FluoReSyn in a stable fashion. These cells were then transiently transfected with well-defined amounts of α Syn expression constructs. Here we could measure easily a dose-response curve for the homogeneously expressed FluoReSyn in the presence of variable amounts of plasmids coding for human α Syn (Fig 1e).

Following the reviewer's comment, we have now also analyzed the correlation between the sensor signal and the immunostaining of α Syn on Reporter-cells (please see Figure 1, below). As expected from our observations and the Western-blot in Fig. 1 of the manuscript, a high correlation between the sensor signal and α Syn amounts was observed.

However, this type of analysis is less trivial to perform in primary hippocampal neurons, as in Figure 6, where we used FluoReSyn to determine if recombinant α Syn is able to enter into the cytoplasm of neurons. These neuronal cultures contain varying levels of the FluoReSyn-expressing plasmid, unlike in Figure 1 of the manuscript, where a homogenous population of stably transfected cells was employed. Additionally, primary hippocampal cultures contain pyramidal neurons of different morphologies, as well as different inhibitory neurons, which enhances the variability of the experimental setup. This is why we did not characterize the system in primary neurons by dose-response curves. This experiment was only performed to determine whether neurons respond to the extracellular application of α Syn. In our opinion, it is therefore expected to encounter a relatively large variation in such an experiment.

This work is the first proof-of-principle of the FluoReSyn sensor, and we are aware that FluoReSyn is not yet working in a straightforward quantitative manner in every cell type, especially if transient transfection is employed. However, it is a tool that can indicate, for example, whether transmittable α Syn exists in human CSF. This is an important type of information, which currently no other technique is able to provide.

Figure 1. Pearson's correlation analysis between the signal from FluoReSyn and immunostaining of α Syn. The experiment was performed using the Reporter cell line transiently transfected with a plasmid encoding for human α Syn. Signals from α Syn immunofluorescence (IF) and FluoReSyn were obtained by analyzing regions of interest (ROI) determined automatically. The ROIs were obtained by identifying all cell nuclei in the DAPI channel, followed by a morphological dilation of the nucleus ROIs, to account for the rest of the cell surfaces. (a) Raw values, in arbitrary units. (b) Double logarithmic graph of the same values. The Pearson's correlation coefficient (r) is indicated in each case.

Reviewer #3:

This is a very interesting paper, with a significant number of strong experiments to support the conclusion that a fluorescent fusion of the NbSyn87 nanobody can serve as a reporter for the presence of human α -Syn in cellular cytoplasm. The range of assays used to measure both target that is over-produced in transfections, as well as penetration of exogenously introduced protein, form a convincing series. Methods descriptions seem very detailed.

We thank the Reviewer for the comments.

The authors have acknowledged in the discussion that there are multiple variables of cell lines, culture conditions, and protocols used to produce α -Syn for the extracellular experiments. It might be worthwhile to explore these questions in a bit more detail, since they are very critical to the interpretation of the results. Not all readers will have the time or access to look up the cited papers. The 7-line final sentence makes it clear that the authors plan to do major optimizations, presumably including much more neuronal cell lines as more faithful reporters. Their more explicit expert opinion on which directions to prioritize could be a useful addition to the field.

We thank the reviewer for this suggestion. We have significantly expanded the explanation on the discussion about the options to produce α Syn *in vitro*, and their implication on the potential variability of the results in the α Syn field. Similarly, we have extended the paragraph discussing potential optimizations of our system and addressed what could be considered relevant as the next steps.

The only experiment that seems to have been incompletely reported is the one testing the human CSF from a range of disorders. While it is clear that the HEK293-derived reporter line is more proof of general concept than definitive, it would still be useful to see the actual distribution of the different diseases in this assay. Since the data are presented individually, it would seem possible to do this, in which case the scatter plot would become a regular, rather than a supplemental, figure.

We did not initially consider to display the details of the patient cohort because this set of patients has not been diagnosed with α Syn-associated diseases. Thus, we probably detect a basal level of transmittable α Syn. Unfortunately, it is even more challenging to obtain CSF from healthy controls, since in Germany this type of sample can typically only be collected from disease cases, so that we cannot state whether such basal levels of transmittable α Syn are also present in healthy donors or young adults.

We have now added the missing details to our graphs (the new Supp. Fig 6). We display the Sensor activity in relation to the clinical diagnoses, as well as to the levels of α Syn present in the CSF of each patient, separated by disease categories. Unfortunately, the small number of patients and the large variability in clinical diagnoses makes it difficult to provide any significant differences between diseases (at least using our cohort). Future work, using larger cohorts and including healthy controls, should test whether this sensor would help in the diagnosis of α Syn-associated diseases.

REVIEWERS' COMMENTS:

Reviewer #1 (Remarks to the Author):

In this revised version and/or in their rebuttal letter, authors meet all comments and critics raised by reviewers.

Reviewer #2 (Remarks to the Author):

Thank you for the clarification re what forms of aSyn the reporter recognizes. This is an important point for use cases of this tool.

I suggest adding the figure provided in the rebuttal letter (relating Fluoresyn readout to IF for aSyn) as a supplemental figure; I found the information valuable, as may others thinking of adopting this tool.

The addition of Supplemental Fig 6 is valuable in interpreting results, even though the cases do not have PD. For both the figure suggested above (relating Fluoresyn readout to IF for aSyn) and Supplemental Fig 6b, please provide both p-value and correlation coefficients for the relationships shown.

Alice Chen-Plotkin

Reviewer #3 (Remarks to the Author):

The changes that were made have clarified several issues, and expanded on others. This increases the value of this paper.

Reviewer #1 (Remarks to the Author):

In this revised version and/or in their rebuttal letter, authors meet all comments and critics raised by reviewers.

We thank the reviewer.

Reviewer #2 (Remarks to the Author):

Thank you for the clarification re what forms of aSyn the reporter recognizes. This is an important point for use cases of this tool.

I suggest adding the figure provided in the rebuttal letter (relating Fluoresyn readout to IF for aSyn) as a supplemental figure; I found the information valuable, as may others thinking of adopting this tool.

This figure is now included in the Supplementary file.

The addition of Supplemental Fig 6 is valuable in interpreting results, even though the cases do not have PD. For both the figure suggested above (relating Fluoresyn readout to IF for aSyn) and Supplemental Fig 6b, please provide both p-value and correlation coefficients for the relationships shown.

Alice Chen-Plotkin

The requested statistics parameters are now included for both Figures in the Supplementary file.

Reviewer #3 (Remarks to the Author):

The changes that were made have clarified several issues, and expanded on others. This increases the value of this paper.

We thank the reviewer.